# scAAV2-Mediated Expression of Thioredoxin 2 and C3 Transferase Prevents Retinal Ganglion Cell Death and Lowers Intraocular Pressure in a Mouse Model of Glaucoma

**DOI:** 10.3390/ijms242216253

**Published:** 2023-11-13

**Authors:** Hee Jong Kim, Seho Cha, Jun-Sub Choi, Joo Yong Lee, Ko Eun Kim, Jin Kwon Kim, Jin Kim, Seo Yun Moon, Steven Hyun Seung Lee, Keerang Park, So-Yoon Won

**Affiliations:** 1Institute of New Drug Development Research, Cdmogen Co., Ltd., Seoul 05855, Republic of Korea; circle-hee@cdmogen.com (H.J.K.); sehocha35@gmail.com (S.C.); mjschoi69@cdmogen.com (J.-S.C.); wlsrnjs407@cdmogen.com (J.K.K.); j.kim@cdmogen.com (J.K.); symoon@cdmogen.com (S.Y.M.); steven@cdmogen.com (S.H.S.L.); keerang.park@cdmogen.com (K.P.); 2Cdmogen Co., Ltd., Cheongju 28577, Republic of Korea; 3Department of Ophthalmology, Asan Medical Center, College of Medicine, University of Ulsan, Seoul 05505, Republic of Korea; jylee.retina@gmail.com (J.Y.L.); csckek@gmail.com (K.E.K.); 4Bio-Medical Institute of Technology, College of Medicine, University of Ulsan, Seoul 05505, Republic of Korea

**Keywords:** glaucoma, thioredoxin 2, *Clostridium botulinum* C3 transferase, intraocular pressure, neuroprotection, oxidative stress, retinal ganglion cells

## Abstract

Elevated intraocular pressure (IOP) in glaucoma causes retinal ganglion cell (RGC) loss and damage to the optic nerve. Although IOP is controlled pharmacologically, no treatment is available to restore retinal and optic nerve function. In this paper, we aimed to develop a novel gene therapy for glaucoma using an AAV2-based thioredoxin 2 (Trx2)-exoenzyme C3 transferase (C3) fusion protein expression vector (scAAV2-Trx2-C3). We evaluated the therapeutic effects of this vector in vitro and in vivo using dexamethasone (DEX)-induced glaucoma models. We found that scAAV2-Trx2-C3-treated HeLa cells had significantly reduced GTP-bound active RhoA and increased phosphor-cofilin Ser3 protein expression levels. scAAV2-Trx2-C3 was also shown to inhibit oxidative stress, fibronectin expression, and alpha-SMA expression in DEX-treated HeLa cells. NeuN immunostaining and TUNEL assay in mouse retinal tissues was performed to evaluate its neuroprotective effect upon RGCs, whereas changes in mouse IOP were monitored via rebound tonometer. The present study showed that scAAV2-Trx2-C3 can protect RGCs from degeneration and reduce IOP in a DEX-induced mouse model of glaucoma, while immunohistochemistry revealed that the expression of fibronectin and alpha-SMA was decreased after the transduction of scAAV2-Trx2-C3 in murine eye tissues. Our results suggest that AAV2-Trx2-C3 modulates the outflow resistance of the trabecular meshwork, protects retinal and other ocular tissues from oxidative damage, and may lead to the development of a gene therapeutic for glaucoma.

## 1. Introduction

Glaucoma is a leading cause of global visual loss. Current estimates suggest that the disease affects up to 80 million people worldwide [1]. Glaucoma is a complex disease involving progressive damage to the optic nerve head, responsible for transmitting visual signals to the brain. The main risk factor for glaucoma is high intraocular pressure (IOP) [2,3]. There are different types of glaucoma, but the most common is primary open-angle glaucoma (POAG), which affects about 70 million people [2,4]. Another type of glaucoma is steroid-induced glaucoma (SIG), which occurs as a result of long-term use of glucocorticoids such as dexamethasone (DEX) [5,6,7]. The mechanism of SIG is believed to be similar to POAG since most POAG patients show an increase in IOP when treated with topical steroids. Both POAG and SIG cause changes in the trabecular meshwork (TM), which is the tissue that regulates the flow of aqueous humor through the eye [8]. Both types of glaucoma result in reduced outflow of aqueous humor and increased IOP, which leads to optic nerve damage and vision loss. Until now, elevated IOP is the only modifiable treatable risk factor. However, even after OP lowering treatments, degeneration of retina ganglion cells (RGCs) and their axonal loss can still progress, ultimately leading to vision loss [9,10]. Thus, there is an urgent need for the treatment to include both effects of IOP-lowering and neuroprotection.

Recently, the Food and Drug Administration approved a new class of glaucoma therapies targeting the actin cytoskeleton [11]. The actin cytoskeleton is a network of protein filaments that provides structural support and shape to the cells. In the eye, the actin cytoskeleton maintains the normal function and homeostasis of the TM [12]. Several studies have shown that alterations in the actin cytoskeleton of TM cells are associated with glaucoma pathogenesis and progression [13]. Fibronectin and alpha-smooth muscle actin (alpha-SMA) are two proteins involved in TM remodeling [11,14,15,16].

Exoenzyme C3 transferase (C3) is isolated from Clostridium botulinum and can inactivate Rho GTPases via ADP-ribosylation [17]. Rho GTPases are upstream signaling molecules of Rho-associated kinase (ROCK). Rho-GTPase signaling is a critical molecular pathway regulating various cellular functions, including cytoskeletal dynamics, cell adhesion, gene expression, and apoptosis [18,19,20,21]. In the eye, Rho-GTPase signaling modulates the contractility of TM cells [22], and increased Rho-GTPase activity, in turn, increases TM contractility and resistance to aqueous humor outflow, thereby increasing IOP and leading to optic nerve damage and vision loss [22,23]. Conversely, Rho-GTPase signaling inhibition reduces TM contractility and lowers IOP [24]. Recent studies have shown that C3 can protect RGCs from glaucomatous injury by inhibiting RhoA signaling and reducing IOP [25]. Therefore, inhibiting the activity of Rho GTPase using C3 may be a promising target for developing novel therapies for glaucoma.

Oxidative stress has been implicated in the pathogenesis of glaucoma, a group of diseases characterized by the progressive loss of RGCs and their axons, resulting in optic nerve head excavation and visual field defects. Oxidative stress can affect various ocular tissues, such as the TM, retina, and optic nerve head while contributing to increased IOP, neuroinflammation, and apoptosis [26,27,28]. These findings suggest that oxidative stress may play a key role in glaucoma pathophysiology and may represent a potential target for novel therapeutic interventions. Thioredoxins (Trxs) [29] are a family of small redox proteins that play important roles in regulating cellular redox homeostasis, protecting cells from oxidative stress, and modulating various signaling pathways. Trx2 is a protein that belongs to the Trx family of proteins, which are involved in various physiological and pathological processes, including antioxidant defense, apoptosis, inflammation, and cancer [29]. Trx2 is an important antioxidant enzyme that responds to oxidative stress in mitochondria. Trx2 has been shown to support RGC survival in experimental glaucoma models by reducing oxidative stress and apoptosis [30,31].

Using fusion proteins as a therapeutic strategy allows the delivery of two or more genes by a single vector structure, which may be advantageous for gene therapy applications [32]. They are useful in biotechnology and medical applications because they target multiple pathways and yield enhanced performance compared to individual therapeutic proteins.

As a singular construct, the virus vector has the potential for increased mobility and survivability in the body while being able to affect multiple therapeutic targets [33]. The regulatory approved and currently-used Aflibercept and Vabysmo, also dual-target fusion proteins, demonstrate the safety and efficacy of fusion protein therapeutics, particularly in the field of ophthalmology [34,35,36].

Thus, we developed a new self-complementary adeno-associated virus 2 (scAAV2) vector that carries a fusion gene of C3 and Trx2 proteins. The scAAV2 is a modified version of the adeno-associated virus 2 (AAV2), a non-pathogenic gene therapy vector. Unlike the conventional AAV2, which requires a second strand synthesis to form a functional genome, scAAV2 has a shorter, self-complementary genome that can fold into a double-stranded structure upon infection. This allows for faster and more efficient transgene expression in the target cells. scAAV2 has been used to deliver therapeutic genes for various diseases, such as hemophilia, retinal degeneration, and Parkinson’s disease [37,38,39].

Gene therapy for glaucoma has several advantages over conventional treatments, such as eye drops or surgery [40]. Gene therapy can provide a long-lasting effect with a single administration, avoiding issues with patient compliance or adverse effects [41]. AAV-based gene therapy has shown promise in ophthalmic conditions such as Leber congenital amaurosis, which received FDA approval for Luxturna in 2017 [42].

In this study, we designed a recombinant AAV vector with the transgene composed of human Trx2 and C3 to treat glaucoma. We show that scAAV-Trx2-C3 efficiently transduces endothelial cells in vitro and in vivo. We offer that intracameral delivery of Trx2-C3 via AAV2 is efficacious at decreasing IOP and preventing RGC cell death in the DEX-induced glaucoma model. We also show that scAAV-Trx2-C3 reduced fibronectin and alpha-SMA expression in glaucoma mouse models. These findings suggest that scAAV-Trx2-C3 is a promising gene therapy vector for glaucoma treatment.

## 2. Results

### 2.1. Characteristics of scAAV2-Trx2-C3

We constructed a self-complementary AAV2 vector containing a fusion gene of Trx2 and C3 transferase (scAAV2-Trx2-C3). The fusion gene was designed with six glycine residues between Trx2 and C3 transferase to preserve their functions as well as to prevent its secretion from the transduced cells without the signal peptide of C3 transferase (Figure 1A). Polymerase chain reaction [28] and western blotting were used to examine the regulated expression of the fusion gene Trx2-C3 in HeLa cells, where we found that scAAV2-Trx2-C3 treatment led to increases in both mRNA (Figure 1B) and protein levels (Figure 1C) of the fusion gene.

### 2.2. scAAV2-Trx2-C3 Inhibits RhoA Activation and Cofilin Phosphorylation at Ser 3

Next, we tested the effects of scAAV2-Trx2-C3 and scAAV2-GFP (control) on RhoA activity, which is known to regulate the contraction of the TM and Schlemm’s canal cells, which together control the outflow of intraocular aqueous humor (AH) and influence IOP. Upon assessing RhoA activity, we found that scAAV2-Trx2-C3-treated cells had markedly inhibited RhoA activity compared to scAAV2-GFP-treated HeLa cells (Figure 2A,B). We then examined the downstream signaling of RhoA, which involves the activation of ROCK1 and ROCK2 kinases and regulates the dynamic organization of the actin cytoskeleton and stress fiber formation. These kinases phosphorylate LIMK, which phosphorylates and inactivates cofilin, an actin-depolymerizing factor. We found that scAAV2-Trx2-C3-treated cells decreased the level of p-cofilin^ser3^ compared to scAAV-GFP-treated cells (Figure 2C,D), suggesting that scAAV2-Trx2-C3 inhibits the RhoA/LIMK-cofilin pathway.

### 2.3. scAAV2-Trx2-C3 Inhibits Oxidative Stress and Fibronectin Expression in Dexamethasone-Treated HeLa Cells

We then performed fibronectin immunocytochemistry using HeLa cells to determine whether scAAV2-Trx2-C3 modulates fibronectin expression and found that scAAV2-GFP/DEX-treated cells (Figure 3B,E,H,J) had increased fibronectin expression levels compared to scAAV2-GFP/PBS-treated cells (Figure 3A,D,G,J). scAAV2-Trx2-C3/DEX-treated cells (Figure 3C,F,I,J) exhibited significantly decreased fibronectin expression levels compared to scAAV2-GFP/DEX-treated cells, demonstrating that scAAV2-Trx2-C3 reduces DEX-induced fibronectin expression.

Next, we evaluated the effects of scAAV2-Trx2-C3 on oxidative stress in DEX-treated cells. Whereas scAAV2-GFP/PBS did not lead to CellROX accumulation (Figure 4A,D,G,J), scAAV2-GFP/DEX-treated cells exhibited increased oxidative stress (Figure 4B,E,H,J), as determined by the accumulation of the fluorescent marker dye CellROX. Meanwhile, compared to scAAV2-GFP/DEX-treated cells, a marked reduction of oxidative stress was observed in scAAV2-Trx2-C3/DEX-treated cells (Figure 4C,F,I,J), with these results suggesting that scAAV2-Trx2-C3 can reduce oxidative damage resulting from dexamethasone treatment.

### 2.4. Intracameral Delivery of scAAV2-Trx2-C3 Lowers IOP in a Dexamethasone-Induced Mouse Model of Glaucoma

To determine the effects of scAAV2-Trx2-C3 on lowering IOP in a steroid-induced ocular hypertension mouse model [5], we utilized DEX induction [43] in 8-week-old C57BL/6 mice. DEX or PBS eye drops were administered twice daily for 8 weeks to induce IOP elevation, as outlined in Figure 5A, and scAAV2-GFP or scAAV2-Trx2-C3 was intracamerally injected into the anterior chamber of the mice (Figure 5A). First, we confirmed the scAAV2-mediated expression of a fusion protein of Trx2 and C3 while also confirming at 2 weeks post-scAAV2-GFP administration the expression of GFP in the TM (Appendix A). In all animals prior to DEX or PBS treatment, the Baseline IOP for both eyes ranged between 12.25 to 13.25 mmHg. At 1 and 2 weeks post-DEX treatment, IOP increased to 21.6 ± 1.9 (scAAV2-GFP) and 20.4 ± 1.7 mmHg (scAAV2-Trx2-C3), respectively (Figure 5B), whereas IOP levels in the scAAV2-GFP/PBS-treated mice (Figure 5B; green line) did not change significantly during the study. Meanwhile, scAAV2-Trx2-C3/DEX-treated mice (Figure 5B; red line) exhibited significantly reduced IOP up to 4 weeks post-injection than scAAV2-GFP/DEX-treated mice (Figure 5B; blue line) at corresponding time points.

### 2.5. Intracameral Delivery of scAAV2-Trx2-C3 Inhibits Fibronectin/Alpha-SMA Expression in the Trabecular Meshwork of the Dexamethasone-Induced Mouse Model of Glaucoma

We examined the expression levels of fibronectin and alpha-SMA in the TM to determine extracellular matrix patterns and alterations of the cytoskeletal structure in DEX-treated mice. Four weeks after scAAV2-Trx2-C3 injection, histological analysis revealed that scAAV2-Trx2-C3/DEX-treated mice (Figure 6B,E,H,J) had significantly reduced fibronectin expression compared to scAAV2-DEX/PBS-treated mice (Figure 6A,D,G,J). We also observed lower alpha-SMA expression in scAAV2-Trx2-C3/DEX-treated mice (Figure 6L,O,R,T) than in scAAV2-GFP/PBS-treated mice (Figure 6M,P,S,T).

### 2.6. Intracameral Delivery of scAAV2-Trx2-C3 Protects Retinal Ganglion Cells in a Dexamethasone-Induced Mice Model of Glaucoma

We conducted a study to evaluate the neuroprotective effects of scAAV2-Trx2-C3 on RGCs in a mouse model of DEX-induced glaucoma. We used immunohistochemistry to measure the number of NeuN-positive cells in the retina. We found that scAAV2-Trx2-C3/DEX-treated mice (Figure 7C,F,I,J) had significantly more NeuN-positive cells than scAAV2-GFP/DEX-treated mice (Figure 7B,E,H,J). Furthermore, we also used the TUNEL assay to detect apoptotic cells in the retina. We observed that scAAV2-GFP/PBS-treated mice had few TUNEL-positive cells in the retina (Figure 7K,N,Q,T) but scAAV2-GFP/DEX-treated mice had markedly increased TUNEL-positive cells in the ganglion cell layer (Figure 7L,O,R,T), compared to which it was shown that scAAV2-Trx2-C3/DEX-treated mice had a reduced number of TUNEL-positive cells (Figure 7M,P,S,T). These results suggest that scAAV2-Trx2-C3 may protect RGCs from DEX-induced apoptosis in mice.

## 3. Discussion

Glaucoma is a chronic, progressive eye disease caused by damage to the optic nerve head, leading to visual field loss [2]. One of the significant risk factors for glaucoma is IOP [3], and even upon undergoing procedures to reduce eye pressure, some patients may yet experience disease progression. Therefore, lowering IOP alone may not be sufficient to prevent or halt the degeneration of the retinal ganglion cells process, a driver of the condition, as other factors, including oxidative stress, inflammation, and excitotoxicity, may contribute to glaucoma pathogenesis [28,44,45,46]. As such, therapeutic approaches for glaucoma, in addition to reducing IOP, should target the disease’s mechanical and biological aspects to confer neuroprotection to the optic nerve [47,48].

In this study, we developed a novel scAAV2 vector carrying a fusion gene of the C3 and Trx2 proteins. The Trx2-C3 transgene encodes a fusion protein of the thioredoxin 2 and C3 transferase, proteins connected by a Gly 6 linker [49,50], which is a widely used and well-established method to link two peptides or proteins. These proteins have distinct therapeutic functions to address different aspects of glaucoma: C3 inhibits the activity of RhoA, which regulates IOP, whereas Trx2 is a potent antioxidant that protects cells from oxidative stress. Combining these proteins into a single fusion gene aimed to achieve a dual effect of lowering IOP while also preventing glaucoma-associated RGC damage. The gene sequences of both proteins do not contain any modifications and are thus identical to their native counterparts. Therefore, they are expected to retain their functions and structures as individual proteins like other dual-target therapeutics used in ophthalmology, such as the fusion proteins aflibercept (Eylea^®^) and faricimab-svoa (Vabysmo^®^) [35,51,52].

Although increased IOP can occur from either the increased production of intraocular fluid or increased resistance to outflow, it is more commonly believed that progressive fibrosis [13] of the TM accounts for most of the damage associated with glaucoma [15,53]. In open-angle glaucoma (OAG), increases in pressure occur because the aqueous humor that fills the eye cannot drain properly through the TM outflow networks [10]. This happens when abnormalities in the TM cells and the ECM they produce lead to TM fibrosis [54,55]. Therefore, regulating the targeting mechanism of TM cells may be an effective therapeutic strategy to address glaucoma.

RhoA, acting as a cytoskeletal protein, regulates the dynamic assembly of the cytoskeleton by activating Rho kinase (ROCK) [23,56], which, alongside LIMK, is how Rho regulates cofilin, with this signal transduction pathway mediating pro-fibrotic processes in many tissues and pathological conditions. Hence, Rho/ROCK pathway inhibitors have been evaluated as anti-fibrotic therapeutics in multiple contexts. Some drugs that block the Rho-ROCK-LIMK-cofilin pathway, which affects the actin cytoskeleton and cell functions, are used to treat glaucoma, a condition with high eye pressure and vision loss. Netarsudil (Rhopressa^®^/Rhokiinsa^®^) was approved in the US and Europe, and ripasudil (Glanatec^®^) was approved in Japan. A new drug, Rocklatan^®^/Roclanda^®^, combines netarsudil with latanoprost, another eye pressure-lowering drug, and has a more substantial effect than the individual components alone [56,57]. In HeLa cells, we first observed that scAAV2-Trx2-C3 significantly inhibits RhoA activity and the phosphorylation of cofilin at Ser 3 before then showing that scAAV2-Trx2-C3 inhibits the increase in fibronectin deposits by DEX both in vitro and in vivo. Therefore, we hypothesized that scAAV2-Trx2-C3 regulates DEX-induced IOP in animal models.

The DEX-induced glaucoma model mimics the clinical features of steroid-induced glaucoma by exposing animals to topical or systemic administration of DEX, a synthetic glucocorticoid. This model allows researchers to investigate the molecular and cellular alterations in the TM and evaluate potential therapeutic strategies to prevent or reverse steroid-induced ocular hypertension and glaucoma. We administered DEX eye drops twice daily to C57BL/6 mice for eight weeks to induce IOP elevation, leading to the accumulation of fibronectin, ECM, and selective RGCs loss by apoptosis, the effects of which resemble human glaucoma pathophysiology.

Here, we show that intracameral delivery of scAAV2-Trx2-C3 lowered IOP in a DEX-induced mouse model of glaucoma. IOP analysis showed that scAAV2-Trx2-C3 significantly decreased IOP levels for up to 4 weeks post-injection when compared to scAAV2-GFP administration in DEX-treated mice. Our study demonstrates that scAAV-Trx2-C3 is a promising candidate for gene therapy to lower IOP, though a limitation of this study is the short-term nature of our observations of the beneficial effects of scAAV-Trx2-C3. However, up to 20% of glaucoma patients show visual field defect progression with RGC and optic nerve degeneration despite the successful management of ocular hypertension. Therefore, there is a need for neuroprotective therapies in glaucoma that can protect the RGCs from damage and enhance their survival and regeneration. Several potential neuroprotective therapeutic modalities, such as neurotrophic factors, antioxidants, and anti-inflammatory drugs, have been investigated in preclinical and clinical studies.

Oxidative stress has been known as a critical factor in ocular hypertension-associated RGC degeneration [58,59,60]. The decreased activity of antioxidant enzymes in the antioxidant defense system, including superoxide dismutase and catalase, has been implicated in RGC death in rat glaucoma models [61,62,63]. With Trx2 being a critical protein in regulating the effects of oxidative stress on RGCs in a rat glaucoma model [64], we demonstrated in vitro that scAAV2-Trx2-C3 can reduce ROS levels in DEX-treated Hela cells. Therefore, we tested whether the scAAV2-Trx2-C3 protects RGCs from DEX-induced oxidative stress upon intracameral delivery in the DEX-induced glaucoma mouse model and found that scAAV2-Trx2-C3 administration prevented the apoptotic cell death of RGCs in vivo. We also found scAAV-Trx2-C3 reduced fibronectin expression and alpha-SMA expression in DEX-induced glaucoma mouse models, further suggesting that it is a promising gene therapy vector against glaucoma.

## 4. Materials and Methods

### 4.1. Antibodies and Chemicals

We used the following primary antibodies in this study: anti-GFP (sc-8334, 200 mg/mL, Santa Cruz Biotechnology, Dallas, TX, USA; IF: 1:100, WB: 1:1000), anti-NeuN (MAB377, 1 mg/mL, MilliporeSigma, Burlington, MA, USA; IF: 1:100), anti-GAPDH (sc-365062, 200 mg/mL, Biotechnology, Dallas, TX, USA; WB: 1:1000), anti-alpha-SMA (ab7817, 1 mg/mL, Abcam, Cambridge, UK; IF: 1:1000), anti-Fibronectin (ab2413, 0.6 mg/mL, Abcam, Cambridge, UK; IF: 1:100), anti-phalloidin (P1951, 300 unit, MilliporeSigma, Burlington, MA, USA; IF: 1:1000), anti-b-actin (sc-47778, 200 mg/mL, Santa Cruz Biotechnology, Dallas, TX, USA; WB: 1:1000), anti-Trx-2 (ab185544, 0.736 mg/mL, Abcam; WB: 1:10,000), anti-RhoA (sc-418, 200 mg/mL, Santa Cruz Biotechnology, Dallas, TX, USA; WB: 1:1000), anti-p-cofilin (CST-3313S, 200 mg/mL, Cell Signaling Technology, Danvers, MA, USA; WB: 1:1000), and cofilin (SC-376476, 200 mg/mL, Santa Cruz Biotechnology, Dallas, TX, USA; WB: 1:1000). We also used Dexamethasone (D1159, MilliporeSigma, Burlington, MA, USA) as a chemical agent.

### 4.2. Animal Care

The 8-week-old male C57BL/6 mice were purchased from Orient Bio (Sungnam, Republic of Korea). All mice were housed in a 12 h light/12 h dark cycle with full access to food and water. All animal care procedures and experiments were conducted according to the Association for Research in Vision and Ophthalmology Resolution on the Use of Animals in Ophthalmic and Vision Research guidelines. The Internal Review Board approved the study for Animal Experiments at the Asan Institute for Life Science (University of Ulsan, College of Medicine).

### 4.3. Construction of Viral Vector

A self-complementary AAV2 vector expressing GFP prepared an expression cassette of the Trx2-C3 fusion gene. For the Trx2-C3 fusion gene, sequences for human thioredoxin 2 (AF480262.1) and C3 transferase (M74038.1) were synthesized (BioD, Gwangmyeong, Republic of Korea), and the synthesized Trx2-C3 gene replaced the GFP gene. The virus vectors used in this study were obtained from Cdmogen Co., Ltd. (Cheongju, Republic of Korea).

### 4.4. Cell Culture

HeLa cells were cultured in Dulbecco’s Modified Eagle Medium (DMEM; Invitrogen, Carlsbad, CA, USA) with 10% fetal bovine serum (FBS, GIBCO, Grand Island, NY, USA), 100 U/mL of penicillin, and 100 μg/mL of streptomycin in a humidified 5% CO_2_ incubator at 37 °C.

### 4.5. Real-Time RT-PCR

scAAV2-EGFP was used as the control, and viral infection was performed at the indicated MOI. For detecting the expression of Trx2-C3 using conventional PCR, HeLa cells were seeded on 12 well plates at 5.0 × 10^4^ cells/well. 72 h after viral infection, total RNA was obtained using Trizol (Thermo Fisher Scientific, Waltham, MA, USA). cDNA was synthesized using M-MLV RT (EBT-1027, Elpis-Biotech, Daejeon, Republic of Korea) and PCR was performed using the StepOne Plus Real-Time PCR System (Thermo Fisher Scientific). The primers used in the study are as follows: Trx2-C3 forward, 5′-TTTAGAGGTGATGACCCTGCTT-3′; Trx2-C3 reverse, 5′- TTGGTCTTCCTCCAAATTGC-3′; β-actin forward, 5′-TGAAGATCAAGATCATTGCTC-3′; β-actin reverse, 5′TGCTTGCTGATCCACATCTG-3′. Protein expression for Trx2-C3 was detected via the standard western blot method, as described. Our experiments verified that no bands were detected in the negative controls (only primers -no template and only template primers). We also did not observe any primer dimers in our gel electrophoresis.

### 4.6. Western Blotting

HeLa cells were seeded on 6 well plates at 2.0 × 10^5^ cells/well and the cells lysed with RIPA buffer (Elpis-Biotech) containing phosphatase inhibitor cocktail 3 (MilliporeSigma) at the indicated time points. Protein concentration was measured using a Pierce BCA Protein Assay Kit (Thermo Fisher Scientific) and equal concentrations of proteins were loaded onto SDS-PAGE gel. After transfer to the PVDF membrane, 5% skim milk in TBST buffer was used for blocking. The membrane was incubated overnight with the primary antibody in a blocking buffer at 4 °C to detect each protein. The membrane was washed with TBST and incubated with a secondary antibody (for rabbit, sc-2357, Santa Cruz; for mouse, sc516102, Santa Cruz) for 1 h. The results were analyzed using a chemiluminescence system (LuminoGraph II, ATTO, Tokyo, Japan) and expression densities were measured using Image J (National Institutes of Health, Bethesda, MD, USA). The experiments were performed at least in triplicate.

### 4.7. Active RhoA Assay

The proportion of active RhoA was detected using an Active Rho Detection Kit (CST-8820S, Cell Signaling Technology). Briefly, HeLa cells were lysed with 1× Lysis/Binding/Wash buffer for 5 min and the supernatant was collected after centrifugation. Part of the supernatant was kept for detecting RhoA as an input control and GST-Rhotekin-RBD and GST-affinity beads were used to precipitate active RhoA. After washing the beads-GST-Rhotekin-RBD-protein conjugate, samples were eluted with a reducing sample buffer. Then, input RhoA and the precipitated active RhoA were detected by a standard western blotting procedure mentioned above using an anti-RhoA antibody.

### 4.8. ROS Measurement

ROS levels in HeLa cells were detected using CellROX Oxidative Stress Reagents (Thermo Fisher Scientific), following the manufacturer’s recommendations. HeLa cells were seeded on 6 well plates at 5.0 × 10^4^ cells/well. After 72 h viral infection, 1 μM of dexamethasone was used for 10 min to treat the cells and induce ROS production, followed by incubation with CellROX for 30 min at 37 °C. The cells were then washed 3 times with PBS and data was obtained using a Nikon Ts2 FL Microscope (Nikon, Tokyo, Japan).

### 4.9. TUNEL Assay

The TUNEL assay used the ApopTag Fluorescein Direct In Situ Apoptosis Detection Kit according to the manufacturer’s protocol (12156792910, Roche Diagnostics, Indianapolis, IN, USA). The retinal sections were then washed 3 times in PBST for 10 min apiece and stained with 4′,6-diamidino-2-phenylindole (DAPI) (Thermo Fisher Scientific).

### 4.10. Immunohistochemistry

The RPE-choroid tissue sections were washed in cold PBS for 15 min and incubated with a universal blocking solution in PBS (0.3% Triton X-100, 1% bovine serum albumin (BSA), 0.05% Tween 20, 0.1% cold fish gelatin, and 0.05% sodium azide) for 1 h at room temperature. The primary antibodies were diluted in an Antibody Diluent (S302283-2, Agilent Technologies, Santa Clara, CA, USA). The samples were incubated with Alexa 488- or Alexa 594-conjugated secondary antibodies for fluorescence microscopy. The nuclei were then counterstained using DAPI.

### 4.11. Intraocular Pressure Measurements and Dexamethasone Treatment of Mice

Mice were anesthetized intraperitoneally with a ketamine/xylazine mix (90 mg/10 mg per kg) and IOP was measured as soon as the mice stopped moving, before the anesthesia effect on IOP occurred, using a rebound TONOVET Plus tonometer (iCare, Vantaa, Finland). The mean was calculated for three IOP measurements for each time point. After baseline IOP was measured, the right eye was treated topically twice daily, 4 h apart, for up to 8 weeks with either one drop (~10 μL) of 0.1% dexamethasone sodium phosphate ophthalmic solution (Bausch & Lomb, Vaughan, ON) or sterile PBS. The contralateral eyes were left untreated. IOP was measured weekly between 4 PM and 5 PM under light conditions.

### 4.12. Intracameral Injection of scAAV2-GFP and scAAV2-Trx2-C3

After induction of ocular hypertension, animals with IOPs of greater than 20 mmHg were selected for grouping. Prior to scAAV2 viral vector administration, mice were anesthetized with Zoletil and Rompun. For topical anesthesia, both eyes received one to two drops of 0.5% proparacaine HCl (Alcaine, Alcon, Geneva, Switzerland). Intracameral injection of the AAV2 viral vectors (1 μL, 5.0 × 107 viral genomes (vg)) into the anterior chamber was performed using a 31-gauge blunted Hamilton syringe. Topical dexamethasone treatment occurred continuously throughout the study.

### 4.13. Statistics and Image Analysis

ImageJ was used to quantify the intensity of the immunofluorescence signals and band densities from immunoblots. The data were analyzed using the Statistical Package for the Social Sciences (SPSS) software (version 11.0, IBM, Armonk, NY, USA). Statistical significance was assessed using an unpaired two-tailed Student’s *t*-test or an ANOVA with Student–Newman–Keuls post hoc analysis. Quantitative data are presented as means ± SEM. Differences were considered significant at *p* < 0.05.

## 5. Conclusions

In summary, we demonstrate here the design of a recombinant AAV vector with a transgene comprised of human Trx2 and C3 to address glaucoma. Our results show that scAAV-Trx2-C3 efficiently transduces endothelial cells both in vitro and in vivo. In contrast, intracameral delivery of Trx2-C3 via AAV2 is efficacious at decreasing IOP and preventing RGC death in a DEX-induced glaucoma mouse model. Additionally, scAAV-Trx2-C3 reduces fibronectin and alpha-SMA expression in mouse models, suggesting its promise as a potential gene therapy vector for glaucoma treatment.

## Figures and Tables

**Figure 1 ijms-24-16253-f001:**
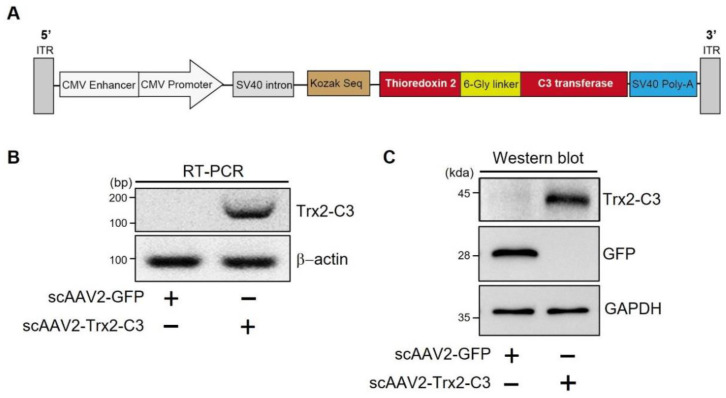
scAAV2-dedicated Trx2-C3 expression in HeLa cells. (**A**) Schematic representation of the viral vector construction. (**B**) RT-RCR for Trx2-C3 mRNA expression at 4 days post-vector transduction at an MOI of 3000. (**C**) Western blot for Trx2-C3 protein expression at 4 days post-vector transduction at an MOI of 3000. ITR, inverted terminal repeat; CMV, cytomegalovirus promoter; SV40, simian vacuolating virus 40.

**Figure 2 ijms-24-16253-f002:**
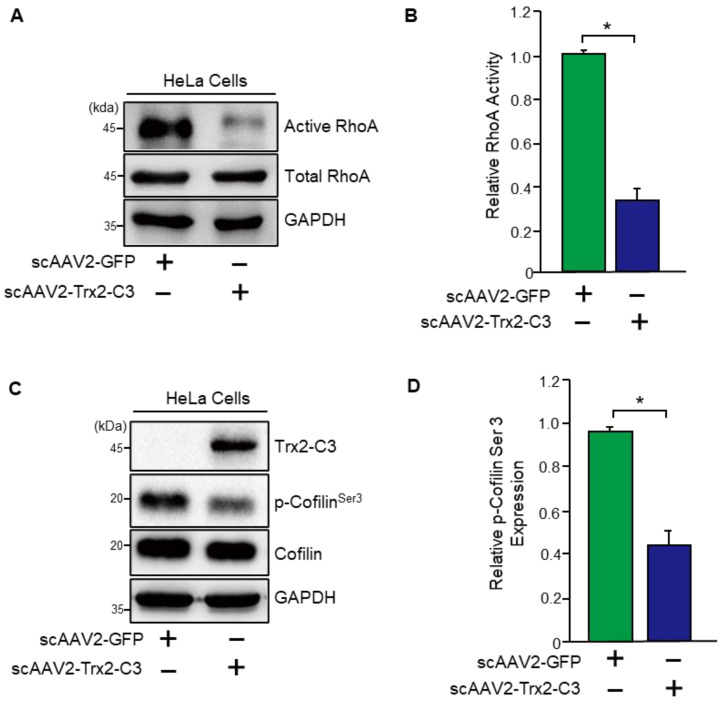
The effect of scAAV2-Trx2-C3 on RhoA activity and the phosphorylation of cofilin^ser3^ in HeLa cells. (**A**) An Active Rho Detection assay kit measured RhoA activity. (**B**) Quantification of the blots shown in (**A**) normalized to total RhoA (n = 3). (**C**) Western blotting for phospho-cofilin (Ser 3), cofilin, Trx2-C3, and GAPDH. (**D**) Quantification of the blots shown in (**C**) normalized to total GAPDH (n = 3). Error bar, mean ± SEM. * *p* < 0.01, unpaired two-tailed Student’s *t*-test. (a.u., arbitrary units). Scale bar, 50 μM.

**Figure 3 ijms-24-16253-f003:**
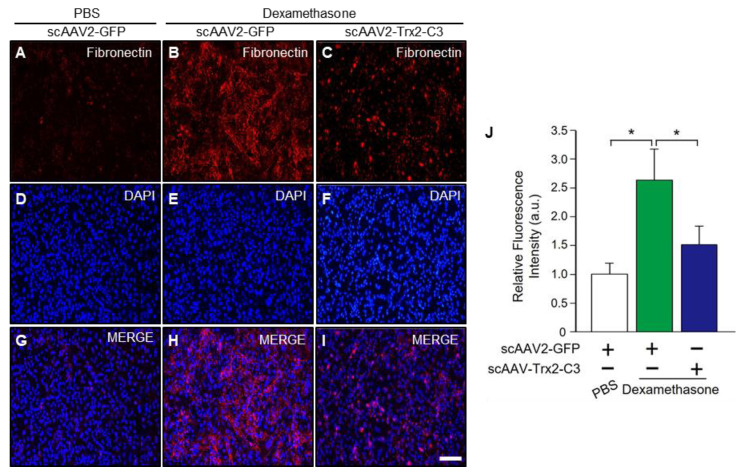
scAAV2-Trx2-C3 reduced fibronectin expression in dexamethasone-treated HeLa cells. (**A**–**I**) Representative images of fibronectin (**A**–**C**; red) immunostaining in HeLa cells treated with PBS or dexamethasone, followed by scAAV2-GFP or scAAV2-Trx2-C3 treatment. Nuclei were counterstained with DAPI (**D**–**F**; blue). Merged images are also shown (**G**–**I**). Scale bar, 50 μM. (**J**) Quantification of fibronectin staining intensity in HeLa cells (a.u., arbitrary units) (n = 3). Error bar, mean ± SEM. * *p* < 0.05, unpaired two-tailed Student’s *t*-test.

**Figure 4 ijms-24-16253-f004:**
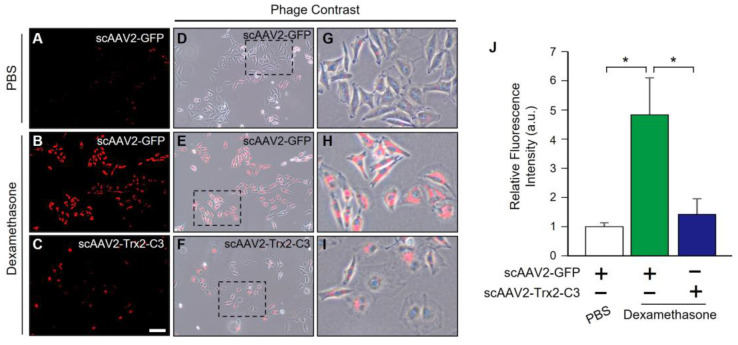
scAAV2-Trx2-C3 reduced oxidative stress in dexamethasone-treated HeLa cells. (**A**–**C**) Representative fluorescent microscopy images of oxidative stress detection in HeLa cells. Scale bar, 25 μM. (**D**–**I**) Images of phage contrast. Comparison of phage contrast, red fluorescence imaging. Boxed areas in the (**D**–**F**) are shown at high magnification as (**G**–**I**), respectively. (**J**) Quantifying oxidative stress in HeLa cells (a.u., arbitrary units) (n = 3). Error bar, mean ± SEM. * *p* < 0.05, unpaired two-tailed Student’s *t*-test.

**Figure 5 ijms-24-16253-f005:**
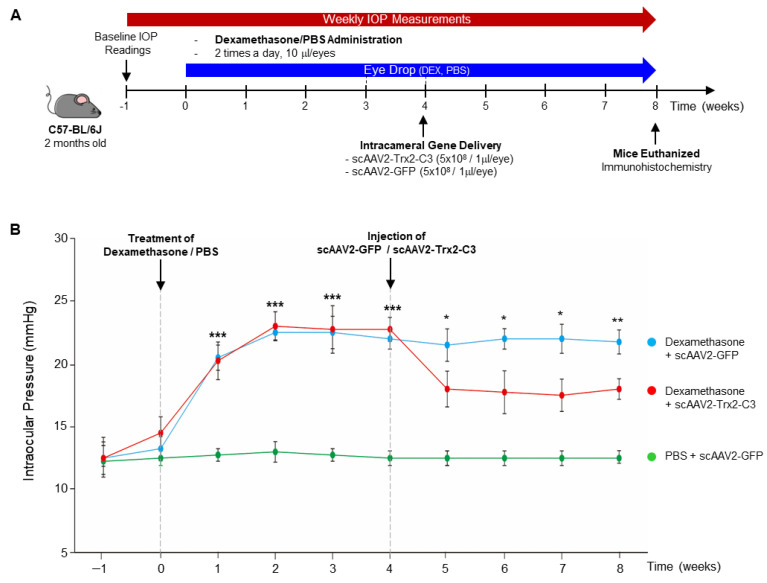
scAAV2-Trx2-C3 reduces IOP in the dexamethasone-induced glaucoma model. (**A**) Experimental scheme. (**B**) The time course of IOP changes in each mouse. Error bars, mean ± SEM. * *p* < 0.05, ** *p* < 0.01, significantly different from scAAV2-Trx2-C3/DEX-treated mice, *** *p* < 0.001, significantly different from scAAV2-GFP/PBS-treated mice (n = 4 per group). ANOVA with Student–Newman–Keuls post hoc analysis.

**Figure 6 ijms-24-16253-f006:**
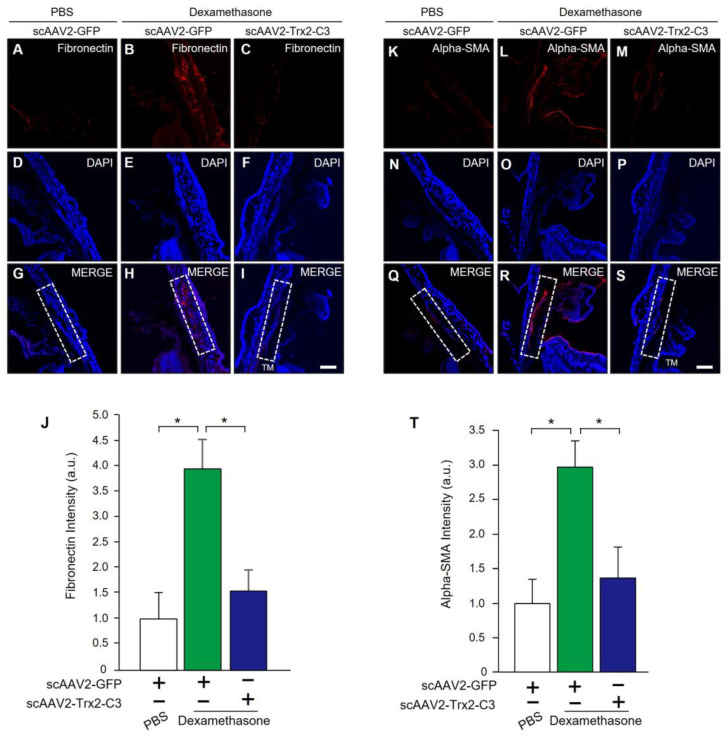
scAAV2-Trx2-C3 reduces fibrotic changes in the TM of the dexamethasone-induced glaucoma model. (**A**–**I**) Representative images of fibronectin immunostaining in the TM of mice injected with PBS or DEX, followed by scAAV2-GFP or scAAV2-Trx2-C3 treatment (**A**–**C**; red). Nuclei were counterstained with DAPI (**D**–**F**; blue). Merged images are also shown (**G**–**I**). Scale bar, 50 μM. (**J**) Quantification of fibronectin staining intensity in HeLa cells (a.u., arbitrary units) (n = 4 per group). (**K**–**S**) Representative images of alpha-SMA immunostaining in the TM of mice injected with PBS or DEX, followed by scAAV2-GFP or scAAV2-Trx2-C3 treatment (**K**–**M**; red). Nuclei were counterstained with DAPI (**N**–**P**; blue). Merged images are also shown (**Q**–**S**). Scale bar, 50 μM. (**T**) Quantification of alpha-SMA staining intensity in HeLa cells (a.u., arbitrary units) (n = 4 per group). Dashed white box indicates a TM area. * *p* < 0.05, ANOVA with Student–Newman–Keuls post hoc analysis (**J**,**T**).

**Figure 7 ijms-24-16253-f007:**
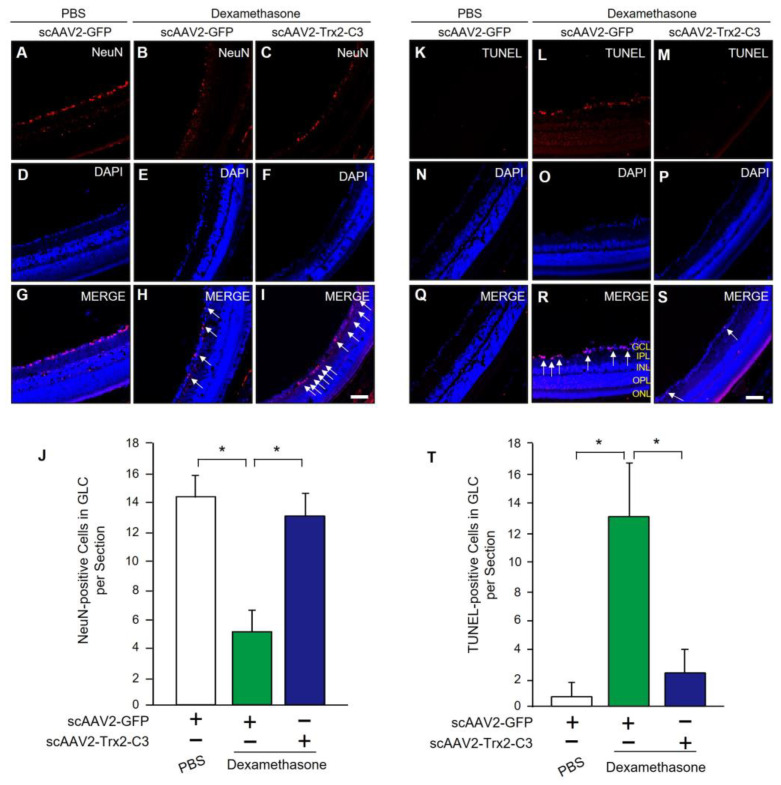
Neuroprotective effect of scAAV2-Trx2-C3 from dexamethasone-induced neurotoxicity. (**A**–**I**) Representative images of NeuN immunostaining in the retinas of mice injected with PBS or DEX, followed by scAAV2-GFP or scAAV2-Trx2-C3 treatment (**A**–**C**; red). Nuclei were counterstained with DAPI (**D**–**F**; blue). Merged images are also shown (**G**–**I**). Scale bar, 50 μM. (**J**) Quantification of NeuN-positive cells (n = 4 per group). (**K**–**S**) Representative images of TUNEL staining (**K**–**M**; red) in the retinas of mice injected with PBS or DEX, followed by scAAV2-GFP or scAAV2-Trx2-C3 treatment. Nuclei were counterstained with DAPI (**N**–**P**; blue). Merged images are also shown (**Q**–**S**). Scale bar, 50 μM. (**T**) Quantification of TUNEL-positive cells (n = 4 per group). White arrows indicate NeuN-positive cells. * *p* < 0.01, ANOVA with Student–Newman–Keuls post hoc analysis (**J**,**T**).

## Data Availability

Data are contained within the article and Appendix A.

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
