# Peer review of "scAAV2-Mediated Expression of Thioredoxin 2 and C3 Transferase Prevents Retinal Ganglion Cell Death and Lowers Intraocular Pressure in a Mouse Model of Glaucoma"

_ijms, 2023, doi:10.3390/ijms242216253_

Round 1

Reviewer 1 Report

Comments and Suggestions for Authors

Dear authors,

While you have picked an important issue to develop novel gene therapy, there are fundamental concerns regarding your experimental strategies.

The rationale of using thioredoxin 2 (Trx2) and exoenzyme C3 transferase (C3) is not clear! Additionally, fusion protein design rationale is not explained either. Therefore, it is very difficult to relate results with conclusion.

I recommend it for major revision. Looking forward to review revised version with more clarity.

Good luck!

Comments on the Quality of English Language

Too many complex sentences.

Reviewer 2 Report

Comments and Suggestions for Authors

This article aimed to develop a novel gene therapy for glaucoma using scAAV2-Trx2-C3 and evaluated the therapeutic effects of this vector in vitro and in vivo using dexamethasone (DEX)-induced glaucoma models. Their results suggest that AAV2-Trx2-C3 modulates the outflow resistance of the trabecular meshwork, protects retinal and other ocular tissues from oxidative damage, and may lead to the development of a gene therapeutic for glaucoma, It is well written and provides novel finding. My only question is scAAV2-Trx2-C3/DEX-treated mice (Figure 5Bexhibited significantly reduced 185 IOP up to 4 weeks post-injection than scAAV2-GFP/DEX-treated mice, How about the effect lasting for? 5 weeks or more?

Round 2

Reviewer 1 Report

Comments and Suggestions for Authors

Dear Authors,

First of all thank you for addressing my first review comments. However, additional issues found and need to be clarified. Please see my detail review comments in the attachment.

Good Luck!

Comments on the Quality of English Language

There are several occasions with typos and too long/complex sentences. I would recommend to use simple sentences instead.

Author Response

Dear Reviewer,

I would like to express my sincere gratitude for the time and effort you have dedicated to reviewing my manuscript. Your insightful comments and suggestions have been incredibly helpful in improving the quality and clarity of our work.

We have carefully considered each of your comments and have made the necessary revisions to our manuscript. We believe these changes have significantly enhanced our manuscript.
Once again, thank you for your valuable contribution to our work. 

Best regards,

So Yoon.
